



# Resolution-dependence of future European soil moisture droughts

Eveline C. van der Linden[1], Reindert J. Haarsma[1], and Gerard van der Schrier[1]

[1]Royal Netherlands Meteorological Institute, De Bilt, Netherlands

*Correspondence to:* Eveline van der Linden (linden@knmi.nl)

**Abstract.** Global climate models project an intensification of future soil moisture droughts over large parts of Europe. This paper investigates the impact of model resolution on the severity and seasonal cycle of future European droughts. We use a 6-member ensemble of the general circulation model EC-Earth to study two periods representative of the start and end of the 21st century under low-to-moderate greenhouse gas forcing (RCP4.5). In our study area, central-western Europe, at high spatial resolution (~25 km) droughts are more severe and start earlier in the season than at standard resolution (~112 km). Here, changes in the large-scale atmospheric circulation and local soil moisture feedbacks lead to enhanced evapotranspiration in spring and reduced precipitation in summer. A more realistic position of the storm track at high model resolution leads to reduced biases in precipitation and temperature in the present-day climatology, which act to amplify evapotranspiration in spring. Furthermore, in the high resolution model a stronger anticyclonic anomaly over the British Isles extends over our study region and supports soil drying. The resulting drier soil induces stronger soil moisture feedbacks that amplify drought conditions in summer. In addition, soil moisture-limited evapotranspiration in summer promotes sensible heating of the boundary layer, which leads to a lower relative humidity with less cloudy conditions, an increase of dry summer days, and more incoming solar radiation. As a result a series of consecutive hot and dry summers appears in the future climate. The enhanced drying at high spatial resolution suggests that future projections of central-western European droughts by CMIP5 models have been potentially underestimated.

## 1 Introduction

Soil moisture drought constitutes a serious hazard to regional European crop yields, water resources, and energy supplies, with major economic impacts (Van Vliet et al., 2015; Blauhut et al., 2016; Stahl et al., 2016). Nevertheless, there remains considerable uncertainty concerning trends in European droughts in response to anthropogenic climate warming (Greve and Seneviratne, 2015). Soil drying is a result of complex interactions between precipitation, water storage in the soil, and evapotranspiration, which are presumably affected by climate warming. Anthropogenic climate change expectedly induces enhanced potential evaporation due to the larger moisture demand of the atmosphere with higher air temperatures (Feng and Fu, 2013; Scheff and Frierson, 2014). Under conditions with sufficient soil moisture, this will lead to an increase in actual evapotranspiration, assuming other variables remain the same. However, atmospheric humidity and wind speed could also change with climate warming and might affect evapotranspiration. At the same time, changes in precipitation are spatially variable between different future projections, leading to enhanced uncertainty at regional spatial scales (Kumar et al., 2013). Moreover,



soil moisture itself interacts with the hydroclimatic components through various feedback processes (Seneviratne et al., 2006, 2010). How these processes will change under climate warming in individual regions is highly uncertain, whereas reliable

regional information is especially important in order to assess future drought risks.

Studying drought changes under climate warming is further complicated by the limited quality and length of observations on historical drought changes. Profile measurements of past soil moisture are sparsely distributed and too short to encompass robust trends (Seneviratne et al., 2010), while satellite-based retrievals are less detailed and only depict up to the top 10 cm of soil (Liu et al., 2012). As an alternative for in situ and remote sensing measurements, reconstructions of soil moisture have been

developed by forcing soil moisture models with historical meteorological data from observations or reanalyses. Using a soil moisture reconstruction, Trnka et al. (2015) revealed a significant soil moisture drying trend for the Czech Republic between the period of 1961–2012 in late spring and early summer. A more frequently used approach to study historical changes in drought is to use various drought indices to quantify drought severity, corresponding to different types of drought (e.g., meteorological, agricultural, or hydrological) and drought characteristics (e.g., severity and duration) (Zargar et al., 2011; Gao et al., 2016).

Owing to the variety of drought indices, assessments of recent dryness trends show large inconsistencies (Seneviratne et al., 2012).

A prominent metric to characterise soil moisture drought is the Palmer Drought Severity Index (PDSI)(Palmer, 1965), which is essentially a simple water balance model. In the PDSI, potential evaporation is often simplified to the parameterisation of Thornthwaite (PDSI_th)(Thornthwaite, 1948), which is only based on temperature and latitude. Sheffield et al. (2012)

showed that the PDSI_th suggests global drying over the past 60 years, but that it may be too simplistic to assess large-scale droughts. They repeated the analysis with the more elaborate Penman-Monteith parameterisation for potential evaporation (PDSI_pm)(Penman, 1948; Monteith, 1965), which indicates that there have been little changes in global droughts over the past 60 years. Moreover, different forcing data of the PDSI_pm index add to the uncertainty of European drought response to climate warming. For example, Trenberth et al. (2014) pointed at high disparities between precipitation datasets used in

the calculation of drought indices. After excluding precipitation datasets with low coverage over land, Dai and Zhao (2017) computed spatial patterns of drying with the self-calibrated version of PDSI_pm (sc_PDSI_pm). They reported a consistent drying trend for southern Europe and a wetting trend for northern Europe between the period of 1950–2012. In addition, they demonstrated that climate warming and the associated vapour pressure deficit over land have enhanced global drying trends since the 1980s.

Regions with consistent increases in observed drought conditions are only partly consistent with climate model simulations from the fifth phase of the Coupled Model Intercomparison Project (CMIP5) (Taylor et al., 2012). Based on the PDSI_pm, model-simulated historical drying appears to be consistent with observation-based estimates over some regions, including southern Europe, but regional changes are still dominated by natural variability (Zhao and Dai, 2017). For the RCP4.5 emission scenario (low-to-moderate greenhouse gas forcing), model projections of PDSI-based drought frequencies show an increase

over nearly all land areas (Zhao and Dai, 2015). Accordingly, the simulated top 10-cm soil moisture content is projected to decline over most of Europe. However, using the same suite of models (CMIP5) but including the entire soil moisture profile in their analyses, Orlowsky and Seneviratne (2013) indicate a large inter-model spread over central-western Europe, with drought





projections ranging from pronounced drying to wetting conditions. Ruosteenoja et al. (2017) show that summer and autumn exhibit strong drying trends in central and western Europe, while all-season drying occurs in southern Europe. This emphasises
the necessity to study seasonal instead of annual mean values for future drying.

Orlowsky and Seneviratne (2013) indicate that by the end of the 21st century, model formulation of global climate models (GCMs) will become the dominant source of spread in drought projections, especially for soil moisture drought and at the regional scale. This suggests that improvements of GCMs could considerably reduce the intermodel-spread in future drought projections. Several studies have shown that high resolution (∼25 km horizontal resolution) GCMs mimic the present-day cli-
mate state more realistically than standard resolution models, since they resolve physical processes more explicitly (Delworth et al., 2012; Jung et al., 2012). Therefore, high resolution simulations are generally expected to be more reliable in their representation of future climatic changes. For example, post-tropical cyclones and mid-latitude storm tracks that bring precipitation towards Europe could change their path in a future climate and are dependent on model resolution (Baatsen et al., 2015; Willison et al., 2015). Furthermore, atmospheric blocking frequencies (Jung et al., 2012; Berckmans et al., 2013) and atmospheric
moisture transport from the ocean to the continents (Demory et al., 2014) are better represented at finer model resolution. Also local-scale land surface-atmosphere feedbacks related to soil moisture are arguably dependent on model resolution (Lorenz et al., 2016). Using high resolution simulations thus could improve our physical understanding and provide potentially better estimates of future changes in drought. This will lead to enhanced confidence in future drought projections at a regional level.

Motivated by these studies, this paper investigates for central-western Europe the consequences of climate warming for
the regional hydrology with a high resolution global climate model. To investigate the role of resolution we have repeated the experiment and analyses at standard CMIP5 resolution for the same climate model. The use of a single high-resolution model allows us to put emphasis on mechanistic explanations for the differences in regional hydrological changes between model resolutions. We assess what the role is of local feedbacks and the large-scale circulation on the response of regional soil moisture storage to climate warming. The study area is central-western Europe, concentrated on a relatively small spatial
scale and therefore riddled with large inter-model differences in future hydrological projections. While most studies focus on annual mean changes, our analysis focusses on warm-season months (April–September). Furthermore, we will analyse changes in the total soil moisture column, which have been shown to exhibit large inter-model uncertainty in future drying over central-western Europe in standard resolution GCMs (Orlowsky and Seneviratne, 2013).

The present paper is outlined as follows. Section 2 presents the data and methods, including a description of droughts def-
initions. Section 3 describes the simulated soil moisture drought characteristics, including climate- and resolution-induced changes in its spatial distribution, seasonal cycle, and severity. The evolution of central-western European droughts is presented in section 4. Here, the land surface water balance components and their interaction with the surface energy balance are examined in detail. Also the effect of changes in the large-scale atmospheric circulation is studied. Thereafter, section 5 shortly describes climatic impacts that are closely related to droughts. Finally, section 6 contains a summary and conclusions.





## 2 Data and methods

### 2.1 Model and experimental setup

This study is based on high-resolution climate model simulations that were also used in previous studies to assess the impact of high resolution on several climatic processes (e.g., Haarsma et al., 2013; Baatsen et al., 2015). The simulations were carried out using the EC-Earth V2.3 model (Hazeleger et al., 2012) at a resolution of T799L91 (∼25 km horizontal resolution, 91 vertical levels), which was the operational resolution at the European Centre for Medium Range Weather Forecasts (ECMWF) from February 2006 until January 2010. Different model runs were performed for the present-day (2002–2006) and future (2094–2098) climate. Each of these datasets consists of a 6-member ensemble spanning 5 years, resulting in a 30-year dataset. In the present-day simulations, observed greenhouse gas and aerosol concentrations were applied, while future concentrations were derived from the RCP4.5 scenario (van Vuuren et al., 2011). Sea surface temperatures (SSTs) were imposed using daily data at 0.25° horizontal resolution from NASA (http://www.ncdc.noaa.gov/oa/climate/research/sst/oi-daily.php) for the 2002–2006 period. The SSTs for the future were calculated by adding the projected ensemble mean change using the 17 members of the coupled climate model ECHAM5/MPI-OM in the ESSENCE project (Sterl et al., 2008) under the SRES A1B emission scenario (Nakicenovic, 2000). This scenario is compatible with the RCP4.5 scenario but the median global temperature increase by the end of the twenty-first century is about 1 °C smaller (Rogelj et al., 2012). Further details on model setup and spin-up procedures can be found in Haarsma et al. (2013) and Baatsen et al. (2015). The simulations were repeated at the standard CMIP5 resolution of T159L62 (∼112 km horizontal resolution, 62 vertical levels) for the same climate model configuration. The generated data have been stored on 5 pressure levels (850, 700, 500, 300 and 200 hPa) at 6-hourly intervals, while surface fields have been saved on a 3-hourly basis.

### 2.2 Land surface scheme

The land surface scheme plays a central role in simulating soil moisture droughts. The land surface scheme used in EC-Earth is HTESSEL (Balsamo et al., 2009). Up to six tiles are present over land (i.e., bare ground, low and high vegetation, intercepted water, and shaded and exposed snow) and two over water (i.e., open and frozen water), with separate energy and water balances. Considering the water balance, precipitation is initially collected in the interception reservoir until it is saturated. Then, excess precipitation is partitioned between surface runoff and infiltration into the soil column. When the imposed water flux exceeds the maximum possible soil infiltration rate, excess water is taken as surface runoff. The vertical discretization of the land surface scheme considers a four-layer soil that can be covered by a single layer of snow. The depths of the soil layers are in an approximate geometric relation (0.07 m for the top layer and then 0.21, 0.72, and 1.89 m for the layers underneath). At both model resolutions all soil and vegetation characteristics of HTESSEL are kept constant. An evaluation of HTESSEL in the Transdanubian region in Hungary by Wipfler et al. (2011) reveals that HTESSEL slightly underpredicts the seasonal evaporative fraction as compared to satellite estimates. The underestimation is most prominent for low evaporative fractions with a maximum prediction error of 30% for individual grid cells.





## 2.3 Observations

To evaluate the EC-Earth simulations we have used the observational E-OBS gridded dataset (0.5° horizontal resolution; version 11.0) for precipitation and 2 m-temperature. The E-OBS data originated from the EU-FP6 project ENSEMBLES (Haylock et al., 2008) and is now maintained and updated by the data providers in the ECA&D project (http://www.ecad.eu). The dataset is based on meteorological station measurements and is designed to provide the best estimate of gridbox averages
to enable direct comparison with climate models. The period used to compare the observational E-OBS data against EC-Earth simulations is 1982–2011.

## 2.4 Drought definitions

Soil moisture droughts, also referred to as agricultural droughts, are characterized by soil water availability. Throughout this paper we quantify the intensity of agricultural droughts by the soil moisture anomaly ($S'$). To compute this drought index, first
the soil water content ($S$) is computed from integrating volumetric soil moisture content ($\theta$) over the total soil profile in the land surface scheme:

$$S = \int d\theta dz \qquad (1)$$

For comparison of present and future climates, the present climatology of soil moisture content averaged over all ensemble members ($S_0$) is used to compute soil moisture anomalies in both the current and future climate. For each ensemble member
and year, anomalies of soil moisture content ($S'$) are computed for each grid point by subtracting $S_0$ from the simulated soil moisture content:

$$S' = S - S_0 \qquad (2)$$

Threshold values for droughts are based on distributions of monthly-mean soil moisture anomalies for the present climate consisting of 30 years of data. They are computed per month, considering each individual year as indepent. Seasonal threshold
values are based on three-monthly averages. For each grid cell, extreme, severe, and moderate droughts are classified to be the 1st, 5th, and 20th percentile of the $S'$ distribution, respectively. With 30 years of data per period, the nearest percentiles that can be calculated are 3.3% (1 out of 30), 6.7% (2/30), and 20% (6/30). As a result, by definition, for the current climate each land surface grid cell is for each individual month on average for 3.3%, 6.7%, and 20% of the ensemble members in the extreme, severe, and moderate drought severity class (s), respectively.

Drought frequency ($f$) is defined as the percentage of months with soil moisture anomalies below a certain drought threshold value. To measure the impact of resolution on the frequency of future droughts we define the resolution factor (RF). RF is defined as the ratio of frequency of droughts in T799 ($f_{\text{T799,s}}$) to T159 ($f_{\text{T159,s}}$) for each drought severity class:

$$\text{RF} = \frac{f_{\text{T799,s}}}{f_{\text{T159,s}}} \qquad (3)$$

Also computed is the corrected resolution factor (CRF) to correct for differences in the present-day climate between T159 and
T799. CRF is defined similarly to RF but uses the threshold values of the T159 present-day climate to compute the frequency





of future droughts at T799 resolution. If CRF and RF are significantly different, this means that differences in the present-day soil moisture climatology could partly explain differences in the frequency of projected droughts.

## 3 Drought characteristics

### 3.1 Spatial distribution and analysis domain

Spring (April–June) and summertime (July–September) European distribution of future soil moisture anomalies for the high
resolution model are shown in Figures 1a and 1b, respectively. These seasonal definitions are slightly different from the common definition and chosen to focus on relevant physical processes that will be discussed later in this paper. Differences in the climate change signal between the high and standard resolution simulations are also depicted (Fig. 1c,d). Both model resolutions simulate strong increases in future spring and summertime droughts over large parts of the Mediterranean, especially in southeastern Europe. Furthermore, in the high resolution model a new drought-prone area emerges that is located over central-
western Europe, roughly coinciding with the area stretching from southwest Germany into Poland and the northern Balkan. This region experiences more severe droughts in future in the high resolution than in the standard resolution simulations. Model resolution seemingly plays an important role for projections of future droughts over central-western Europe.

This motivates our choice for the study region in central-western Europe, defined as the area between 47–52°N and 5–10°E (black box in Fig. 1), which roughly encompasses the Rhine-Meuse drainage basin and partly overlaps with the region studied
by Van Haren et al. (2015a). The choice for this region is further motivated by the importance of this area for the fresh water supply to the Netherlands and Belgium, which are situated in the lower part of the Rhine-Meuse delta. Consumable water, shipping, irrigation, and cooling of power plants all depend on sufficient water supply from the Rhine-Meuse basin and could be affected by severe droughts.

### 3.2 Seasonal cycle

Soil moisture content over central-western Europe exhibits a large annual cycle and reaches its minimum value in September (Fig. 2a). Note that these values depend on the soil depth of the land surface scheme, and therefore are model dependent. For the present-day (baseline) climates, there are no significant resolution-dependent differences in monthly soil moisture content values, except for March. The projected changes in soil moisture content are, however, strongly season and resolution-dependent. At both resolutions, the amplitude of the annual cycle is larger in the future climate than in the present-day climate
due to drier soils around the annual soil moisture minimum.

Figure 3 depicts monthly volumetric soil moisture anomalies over the four soil layers of HTESSEL under 2100 conditions, with respect to the current climate. At standard resolution, future soil moisture reduction is mainly confined to late summer and autumn. An almost full recovery of soil moisture occurs in late winter for top soil layers, and in late spring for deeper layers. On the other hand, high resolution simulations exhibit year-round soil moisture depletions compared to the climate at the start
of the 21st century. At high resolution, the soil moisture content over the top three soil layers of HTESSEL restores to its





former state in late winter, but the soil in the fourth layer remains drier than current values, suggesting a year-round reduction of ground water. The reduced soil water also impacts year-round runoff values (Fig. 2b).

As a result, in late spring (April–June), future soil moisture content (Fig. 2a) differs significantly from present-day values at high model resolution ($S' = -45$ mm) whereas there is no significant change at standard model resolution ($S' = -6$ mm). At the start of summer, the soil is thus preconditioned towards drier conditions in T799 than in T159. Furthermore, future

late summer (July–September) soil moisture depletion is on average more severe in T799 ($S' = -90$ mm) than in T159 ($S' = -50$ mm) (Welch t-test: p = 0.01). The more severe soil dessication in late spring at high resolution is thus maintained throughout the late summer months. In the rest of the paper we therefore focus on the modelled differences in late spring (preconditioning phase) and late summer (maintenance phase). For brevity, from now on we will refer to 'spring' for the period April to June and 'summer' for the period July to September.

## 3.3  Severity

The changes in the seasonal cycle of soil moisture content affect the severity of future droughts under climate warming. Consider Figure 4, which depicts the $S'$ distributions of spring and summer at T799 and T159 resolution, for the present and future climates. The drought threshold values as defined in section 2.4, which are based on these distributions, are also indicated. The associated frequency of extreme, severe, and moderate droughts by the end of the twenty-first century, averaged

over central-western Europe, can be seen in the bar charts of Figure 5.

In spring, model resolution has a high impact on the frequency of extreme, severe, and moderate droughts. In the T159 simulations, extreme future springtime droughts occur about once every thirty years over central-western Europe ($F_{\text{T159,extreme}} \simeq 3\%$) (Fig. 5), which is comparable with the current climate. There is also no significant change in the frequency of springtime severe and moderate droughts in T159. This is clearly visible in the large overlap of the present-day and future climates in the

distributions of spring soil moisture anomalies for T159 (Fig. 4a). In contrast, in T799, the springtime soil moisture anomaly distributions indicate that the frequency of extreme droughts increases due to a shift towards drier conditions in a future climate. Future extreme springtime droughts are about seven times more frequent in T799 than in T159 ($RF_{\text{extreme}} \simeq 7.3$) and occur about once every four years ($f_{\text{T799,extreme}} \simeq 24\%$)(Fig. 5). Note here that the threshold value for T799 extreme springtime droughts (red line) is more or less similar to the threshold value in T159 due to the similarity of the baseline soil moisture

distributions. Also, the frequency of severe ($f_{\text{T799,severe}} \simeq 41\%$) and moderate ($f_{\text{T799,severe}} \simeq 71\%$) droughts increases in the high resolution simulations. This is partly due to the increase in extreme droughts, which are by definition included within the threshold values of moderate and severe droughts.

During the summer months, future extreme droughts become more frequent at both resolutions ($f_{\text{T799,extreme}} \simeq 71\%$ and $f_{\text{T159,extreme}} \simeq 21\%$). Nevertheless, model resolution is still an important factor in drought frequency with over three times more

extreme drought months in T799 than in T159 ($RF_{\text{extreme}} \simeq 3.4$). The increase in the number of extreme summertime drought months is partly due to a shift of the threshold value of $S'$: in T159 extreme droughts are defined as $S' \leq -99$ mm compared to $S' \leq -60$ mm in T799. This is due to a relatively narrow distribution of soil moisture anomalies in the high resolution baseline climate (Fig. 4d). Applying the threshold of T159 to T799 results in 49% of the months being under extreme drought conditions,





still more than a doubling in the number of months in extreme drought compared to T159 simulations ($CRF_{extreme} \simeq 2.3$). The resolution factor becomes less pronounced for summertime severe ($RF_{severe} = 2.6$) and moderate ($RF_{moderate} = 1.5$) droughts. This is because the increase in moderate and severe droughts in summer is mainly due to the increase in extreme droughts (Fig. 5). In other words, the high resolution simulations indicate that extreme droughts in the present-day climate will become the normal climatological state in the future.

## 4  Drought evolution

Under similar soil and vegetation properties, changes in soil moisture content depend largely on precipitation and the evaporative demand of the atmosphere. An accurate simulation of precipitation and evapotranspiration in the current climate would increase our confidence in the model performance of future drought projections. Therefore, in subsection 4.1 we first compare the simulated climate to the E-OBS observations. After the model validation we study the land surface water balance in subsection 4.2, which includes precipitation and evapotranspiration as important factors. The water balance provides more information on the drivers of soil moisture droughts. In the last two subsections, the focus is on mechanistic explanations for changes in the land surface water balance. These explanations are separately discussed for spring in subsection 4.3 and summer in subsection 4.4. The discussion includes local feedback mechanisms as well as large-scale processes.

### 4.1  Validation of present-day climate

To gain some confidence in the model performance we validate the present-day climate over central-western Europe. This is done for simulated present-day precipitation and near-surface air temperature. The latter is used instead of evapotranspiration since observations of evapotranspiration are scarce and air temperature mainly determines the evaporative demand of the atmosphere. Figure 6a and 6b show the annual cycle of precipitation and near-surface air temperature over the analysis domain for E-OBS observations and the EC-Earth model simulations at high and standard resolution.

Precipitation biases of present-day climate simulations show that it is too wet over the study region (Fig. 6c). As discussed in Van Haren et al. (2015b) this is due to a too zonal position of the storm track over the east Atlantic Ocean and Western Europe. This is a common bias in CMIP3 and CMIP5 models (Haarsma et al., 2013; Van Ulden et al., 2007). The wet bias is reduced in the high resolution version of EC-Earth primarily due to a less zonal orientation of the storm track (Van Haren et al., 2015b).

Surface air temperature biases show that the T159 version is a bit too cold compared to observations (Fig. 6d), whereas T799 is somewhat warmer, especially in summer. The warmer temperature is associated with the less zonal orientation of the storm track which makes the summer climate less maritime and more continental. The high resolution simulations thus perform better with respect to the E-OBS precipitation and temperature data than the standard resolution model. We shall see later that this contributes to enhanced future drying at high resolution.



## 4.2 Land surface water balance

The land surface water balance gives more information on the drivers of soil water changes:

$$dS/dt = P - ET - Q \qquad (4)$$

Here, soil moisture storage (dS/dt) is the change in soil moisture content over time, P is precipitation, ET is evapotranspiration and $Q$ is runoff. To study climate-induced changes in more detail, the land surface water balance can also be written in terms of anomalies, linking storage anomalies (dS'/dt) to flux anomalies (P', ET', Q'). In our approach anomalies are (future) changes relative to the present-day climatology. Figure 7 shows on the left the average storage anomalies in a future climate compared to the current climate from April to September. On the right the contributions of P', ET', and Q' to the changes in dS'/dt are depicted.

For both model resolutions, the anomalies of evapotranspiration and precipitation clearly show that climate change induces enhanced evapotranspiration in spring and reduced precipitation from June to September. Throughout the entire warm season, the reduced future soil water content yields decreases in runoff, leading to reduced availability of water resources downstream. At the same time, the decreases in runoff restrict the local soil moisture storage anomalies, preventing the soil from further desiccation (Fig. 7).

In the following subsections, we focus on spring and summer averages for mechanistic explanations of the simulated changes in the surface water balance. Individual months are discussed if it adds relevant information to the analyses.

## 4.3 Spring evapotranspiration increase

The largest resolution-induced differences in soil moisture drying occur in spring (Fig. 7). Summed over spring months, drying is larger in T799 (dS'/dt = –30.9 mm) than in T159 (dS'/dt = –11.1 mm). Compared to the present, the future shows an intensification of the hydrological cycle with enhanced evapotranspiration from April to June and enhanced precipitation in April and May (Fig. 7). At both model resolutions, evapotranspiration increase is the dominant contributor to spring soil drying (Fig. 7). This increase is stronger in T799 (ET' = +55.4 mm) than in T159 (ET' = +21.6 mm). Contributions of average springtime precipitation to drying are negligible since rainfall increases in April and May are compensated by a rainfall reduction in June. Furthermore, runoff plays an essential role in the magnitude of future changes of the land surface water balance. Both model resolutions simulate reduced future runoff, indicating that less water flows away from the study region. Reduced runoff thus counteracts soil drying in central-western Europe (Fig. 7), an effect which is stronger at high model resolution. However, the relative 'moistening' effect (positive value of dS'/dt) of reduced runoff does not fully compensate for the strong drying effect of enhanced evapotranspiration in T799. The net result is stronger spring drying in central-western Europe at high model resolution. In the remainder of this subsection, we focus on the dominant contributor to future spring drying, which is evapotranspiration.

Since local moisture recycling plays an important role in evapotranspiration over the continents, first we assess resolution differences in evapotranspiration in the present-day climate. The annual cycle of evapotranspiration over central-western Europe





(Fig. 2d) indicates that T799 simulates lower evapotranspiration in the present-day climate than T159. As already discussed in subsection 4.1, the precipitation bias over central-western Europe is smaller in T799 than in T159 (Fig. 6b). In spring, the high resolution model simulates significantly less precipitation (2.9 mm/day) than the standard resolution model (3.4 mm/day) (Welch t-test: $p < 0.01$). This is induced by a shift of the precipitation distribution towards more dry days ($P \leq 1$ mm day$^{-1}$) in the high resolution model compared to the standard resolution model (Kolmogorov-Smirnov test: $p < 0.01$). The reduced

precipitation likely induces less evapotranspiration in T799 (2.9 mm/day) than in T159 (3.3 mm/day) in the current climate (Welch t-test: $p < 0.01$). In other words, suppressed precipitation in the present-day climate might lead to reduced recycling of moisture over land. This is because in HTESSEL rain is initially collected in the interception reservoir. Before saturation of the interception layer, less rainfall results in less evapotranspiration, which could explain the reduced evapotranspiration in spring in the current climate.

Second, we assess what mechanisms could explain the enhanced increase in evapotranspiration over the 21st century. Assuming that the top soil layer is still saturated at the start of spring, this implies that the enhanced evapotranspiration in T799 is atmosphere-driven. We start with examining future changes in the atmospheric circulation. In the high resolution model an anomolous high pressure area develops near the British Isles that extends over central-western Europe (Fig. 8). The results show that the anomalous anticyclonic circulation does not significantly affect the wind climate over central-western Europe

(Fig. 9a). However, high pressure anomalies promote subsidence, thereby supporting clear skies with more solar radiation reaching the surface (Fig. 9b). This effect is visible in the enhanced solar radiation in June, but not in April and May. As a consequence, more energy is available for future evapotranspiration and surface warming (Fig. 9c) in June in T799.

    Third the impact of air temperature on atmospheric humidity is examined. Present-day daily mean springtime temperatures are higher in T799 (12.3 °C) than in T159 (11.8 °C) (Welch t-test: $p < 0.01$) (not shown). These warmer springtime temperatures

in T799 are associated with 10% lower cloudiness and 12 W m$^{-2}$ more surface solar radiation in the baseline climate than in T159. The reduced cloudiness is consistent with the smaller amount of precipitation in the T799 model. In the future climate, these resolution-dependent differences are roughly maintained (9% reduced cloud cover and 15 W m$^{-2}$ more surface solar radiation than in T159). The specific humidity (q) of the atmosphere increases with climate warming (Fig. 9d). However, as temperature increases, the increase of saturation vapour pressure is larger in a warmer climate due to the nonlinearity of the

Clausius-Clapeyron relation. The warmer baseline surface air temperature in T799 leads to a stronger increase in saturation vapour pressure for a similar daily mean temperature change (+2.7 °C in T799 compared to +2.3 °C in T159) and thus a larger increase in the atmospheric demand for moisture. This is clearly visible in the lower relative humidity of the atmosphere at high model resolution (Fig. 9d). Higher baseline surface air temperatures thus contribute to enhanced springtime evapotranspiration in T799 compared to T159 under future climate warming.

In summary, spring drying is larger in the high resolution model version. This is due to a more continental climate in the high resolution model with less precipitation and warmer near-surface air temperatures, which promote enhanced evapotranspiration under climate warming. The main cause of the more continental climate is the less zonal position of the storm track in T799 as explained in Van Haren et al. (2015b).



### 4.4 Summer precipitation decline

Summer drying anomalies are larger than spring anomalies, but the summer differences between the model versions are relatively small. Compared to the present, the future increase of summer drying at T799 resolution (dS'/dt = –48.1 mm) is only slightly larger than the increase at T159 resolution (dS'/dt = –44.1 mm). Consequently, the resolution-induced differences in soil moisture content developed in spring are more or less maintained during summer. The dominant contributor to future soil

drying in summer is a decline in rainfall (Fig. 7). Consistent with enhanced drying, the decline in total summer precipitation is larger in T799 (P' = –72.3 mm) than in T159 (P' = –59.6 mm). Similar to spring months, runoff acts as a negative feedback on soil drying. Reduced runoff counteracts local soil drying in both T799 (Q' = –21.5 mm) and T159 (Q' = –12.2 mm). However, runoff is rather a result of the reduced water availability in the soil than an active contributor to the drying. Therefore we focus on the causes of the precipitation decline.

We first study the large-scale changes in the atmospheric circulation. During summer, an anomolous high pressure area over the British Isles extends over central-western Europe in both model versions (Fig. 10). The anticyclonic circulation anomaly promotes subsidence and decreases convection. This leads to less clouds and reduced summer precipitation (Fig. 9b and 7). In addition, changes in the atmospheric circulation potentially affect horizontal moisture transport through changes in the wind climate over central-western Europe (Fig. 9a). Over this region, eastward winds (U10) typically bring oceanic moisture over

land which increases the likelihood of continental precipitation. Due to the anticyclonic circulation anomaly, eastward winds are projected to decrease significantly towards the future at both T159 (U10' = –0.50 m s$^{-1}$) resolution and T799 resolution (U10' = –0.30 m s$^{-1}$) (Fig. 9g), reducing advection of atmospheric moisture from ocean sources. Zonal wind speed changes thus act to enhance summer soil drying over central-western Europe. However, zonal wind speed changes do not induce the resolution-dependent differences in future summer drying, since wind speed reductions in T799 are smaller than in T159 and

thus have a counteractive effect on the differences in drying.

Furthermore, soil drying and precipitation decline anomalies are not uniformly spread over individual summer months. In August and September, soil moisture storage anomalies are larger in T799 than in T159 (Fig. 7). Consistently, in August and September rainfall decline is larger in T799 than in T159 (Fig. 7). Remarkably, in July the climate change-induced soil moisture drying and precipitation decreases are larger in T159 than in T799 (Fig. 7). The simulations indicate that these

monthly precipitation anomalies are largely determined by the magnitude and location of the anomolous high pressure pattern over the British Isles. In August and September, the anomolous high pressure is stronger over central-western Europe in T799 (Fig. 11b–c), whereas in July, the high pressure anomaly is larger in T159. It thus appears that the atmospheric circulation plays an important role in monthly precipitation changes, but is not the main cause of resolution-induced differences averaged over the summer months.

Next to large-scale circulation changes, local feedbacks play a role in the reduced precipitation in summer. The surface water balance is closely connected to the surface energy balance through latent cooling by evapotranspiration. Figure 12 depicts future anomalies of the main components of the surface energy balance, including the sensible heat flux (SH'), latent heat flux (λET'), and the net surface radiation (R$_{net}$'). In future, maximum evapotranspiration is achieved in June for both T159 and





T799 (Fig. 2). Thereafter evapotranspiration decreases rapidly, coinciding with a significant decrease of soil moisture in the top layer (Fig. 3). This decrease is stronger in T799 than in T159 (Fig. 8) and shows that from June onward evapotranspiration is limited by soil moisture. In the present-day climate, maximum evapotranspiration occurs one month later (July), for both T159 and T799, closer to the maxima in solar radiation and surface temperature (Fig. 9b,c), indicating that for the present climate evapotranspiration is not restricted by soil moisture but by available surface energy. As a result, there is no marked

change in total summer evapotranspiration towards the future in T799 (ET' = +3.9 mm) and T159 (ET' = 4.1 mm). In turn, the minor increase in evapotranspiration limits the increase in specific humidity of the atmosphere (Fig. 9d) and thereby the local recycling of moisture. In addition, the limited atmospheric moisture content in combination with warmer temperatures leads to reduced relative humidity and less cloudiness with more surface solar radiation (Fig. 9b,d). The resulting warmer and drier boundary layer (Fig. 9c,d) further diminishes rainfall. This effect is stronger at T799 resolution, presumably caused by

the drier soils at the start of summer that alter the partitioning of the turbulent fluxes. Local feedbacks of soil moisture with temperature and precipitation thus seem to dominate the resolution-dependent differences in summer drying.

## 5 Drought impacts

### 5.1 Warm extremes

The drier soil at the start of summer limits the evaporative fraction leading to enhanced sensible heat fluxes that amplify

near-surface warming (Fig. 9 and Fig. 10). Combined with warmer air temperatures, this decreases the relative humidity of the atmosphere (Fig. 9d). The warmer and drier boundary layer inhibits cloud formation and further enhances surface solar radiation (Fig. 9b). The resulting rise in daily maximum surface air temperature in T799 (SAT' = +5.1 °C) and T159 (SAT' = +4.8 °C) is about two times larger than in spring.

This local soil moisture-temperature feedback is starts a few weeks earlier (Fig. 9b) for the high resolution model due to the

enhanced soil drying in spring. The strenghtening of the soil moisture-temperature coupling is also visible in Figure 12, which shows a steeper regression line between soil moisture content and daily maximum temperature for the future climate than for the present climate. The stronger feedback in T799 leads to a clustering of exceptional dry and hot summers in T799 compared to T159 (clustered in the top left corner in Fig. 12b). The strong link between soil moisture and maximum temperatures is created by reduced soil moisture which strongly restrains latent cooling in summer in both T159 (r = 0.78, p < 0.01) and T799

(r = 0.90, p < 0.01). In the future, drier future spring soil conditions therefore could potentially act as a precursor for hot summers (e.g., Rasmijn et al., 2018).

### 5.2 Heavy precipitation and dry days

In the current climate, T799 simulates a larger fraction of dry days and a larger fraction of precipitation falling as intense rain compared to T159 (Fig. 13), which is presumably caused by the higher surface air temperature. In the future climate, both

resolutions simulate an increase in the number of dry days in summer and in the contribution of heavy precipitation to the



total precipitation amount. This effect is larger in T799. Unless there is moisture supply from the ocean, after a day with heavy precipitation the air needs to be recharged before a new downpour can start. In between heavy precipitation days, it is therefore relatively dry with little rain. During these dry days there will be less clouds, with more incoming solar radiation, higher temperatures, and less relative humidity inducing a higher evaporative demand. So for the same amount of rain, a shift to more dry days and heavy precipitation potentially increases the evaporative demand of the atmosphere. Dry soils, however,

limit the rate of evapotranspiration and slow down the rate at which the atmosphere can recharge with water vapour. The drier soil conditions in T799 thus likely increase the number of dry days between heavy rain events.

## 6   Summary

Using two model resolutions of the global climate model EC-Earth we have investigated the impact of model resolution on the occurrence and causes of extreme droughts in the Rhine-Meuse drainage area in central-western Europe. For the present-

day climate the high resolution (T799, ~25 km) and standard resolution (T159, ~112 km) version have similar soil moisture climatologies. However, for the future large differences occur. In spring, enhanced evapotranspiration with respect to the corresponding baseline climatology induces much stronger drying in the high resolution model, which leads to an sevenfold increase in the frequency of extreme soil moisture droughts. The enhanced spring drying preconditions the soil to a drier state at the start of summer. Both resolutions simulate a significant increase in summer drying of similar magnitude. Due

to the predrying in spring, this leads to over three times more frequent extreme droughts at high resolution than at standard resolution. Consequently, in the high resolution model, extreme summer droughts in the present-day climate become the normal climatological state in the future.

The analyses revealed that both anomalous circulation patterns and local feedbacks are responsible for the increase in the frequency and severity of droughts in a future climate. A less zonal position of the storm tracks leads to a more continental

climate over central-western Europe and a reduction of present-day biases in precipitation and surface air temperature. The resulting warmer and drier spring promotes enhanced evapotranspiration in the high resolution model. Furthermore, an anticyclonic pressure anomaly over the British Isles extends over central-western Europe. Although no significant changes in wind occur, the high pressure anomaly supports clear skies with enhanced surface solar radiation reaching the surface. This provides additional surface energy for turbulent fluxes in June.

Summer drought characteristics are enhanced by a reduction in rainfall. The drying of the ground results in a reduced evaporative fraction and combined with enhanced solar radiation in higher surface temperatures. This feedback is stronger at high model resolution due to the predrying in spring. The drier soil in the high resolution model at the start of summer induces stronger local feedbacks with precipitation and evapotranspiration, which leads to a clustering of extreme droughts and warm extremes. The impact of soil moisture on observed European warm extremes has also been demonstrated by Whan et al.

(2015) using a simple water balance model. Another consequence of the occurrence and clustering of extreme droughts is the absence of recovery of the ground water storage in the following winter with consequences for year-round runoff. The stronger



local feedbacks open on the other hand possibilities for predictability of summer droughts, heat waves, precipitation, and river runoff.

We argue that both dynamical and local feedback processes are better represented at higher resolution. The cause of the anomalous anticyclonic circulation over the British Isles could be a weakening of the Atlantic meridional overturning circulation (AMOC) under future climate warming as discussed in Haarsma et al. (2015). The stronger response at the high T799 resolution is probably due to enhanced ocean-atmosphere coupling. Because CMIP5-models underestimate the natural inter-annual variability of this AMOC-induced UK high (Haarsma et al., 2015), we argue that the stronger response in T799 is more realistic. Furthermore, the additional increase of spring droughts in the high resolution simulations is partly due to a better representation of the Atlantic storm track at high model resolution as described in Van Haren et al. (2015b). Whether the simulated droughts and the impact of resolution is robust has to be evaluated in a larger ensemble of different models. Initiatives as the Horizon 2020 project PRIMAVERA (https://www.primavera-h2020.eu/) and the CMIP6-endorsed HighResMIP (Haarsma et al., 2016) will contribute to the hypothesis put forward in this study that the simulated European droughts by CMIP5 are underestimated due to dynamical causes and local feedbacks that can only reliably be simulated at high resolution.

*Competing interests.* No competing interests are present.



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





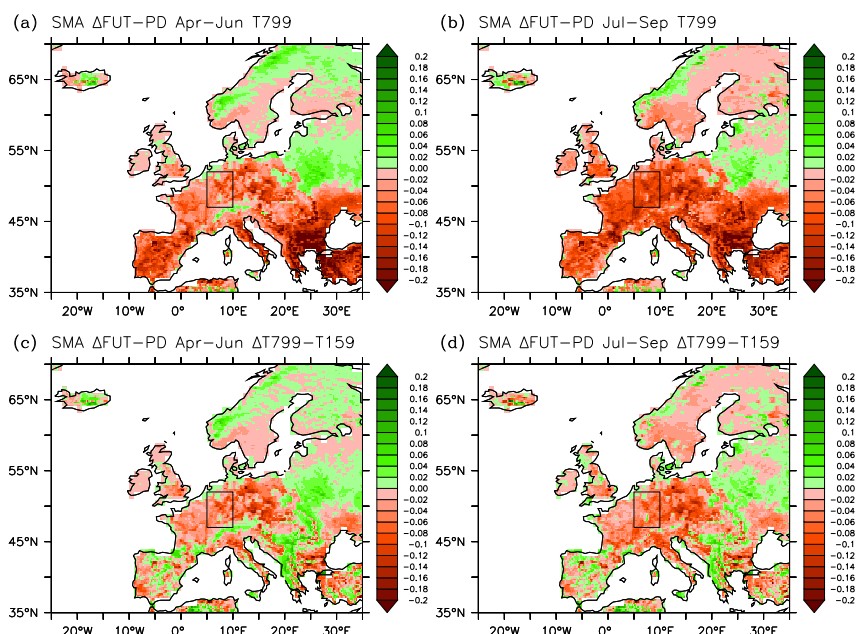

**Figure 1.** Top: Soil moisture anomaly climate change signal (future – present) for (a) spring and (b) summer in the high resolution model.

Bottom: (c–d) Same but for difference in climate change signal between the high and standard resolution model.

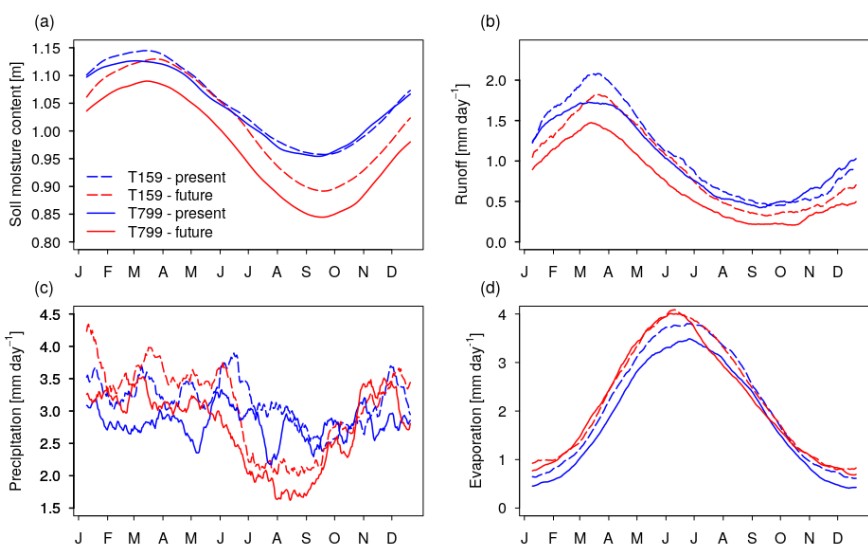

**Figure 2.** Annual cycle of (a) soil moisture content, (b) runoff, (c) precipitation, and (d) actual evapotranspiration. Values are smoothed with a 20-day running mean filter and averaged over central-western Europe. The letters on the horizontal axis correspond with the first day of each month (based on a year with 365 days).



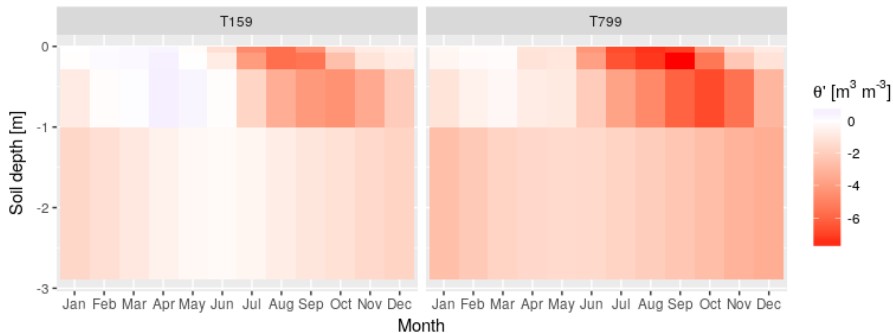

**Figure 3.** Profile of volumetric soil moisture monthly anomalies ($\theta$') under 2100 conditions with respect to the present-day climatology over central-western Europe for standard and high spatial resolution.

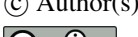


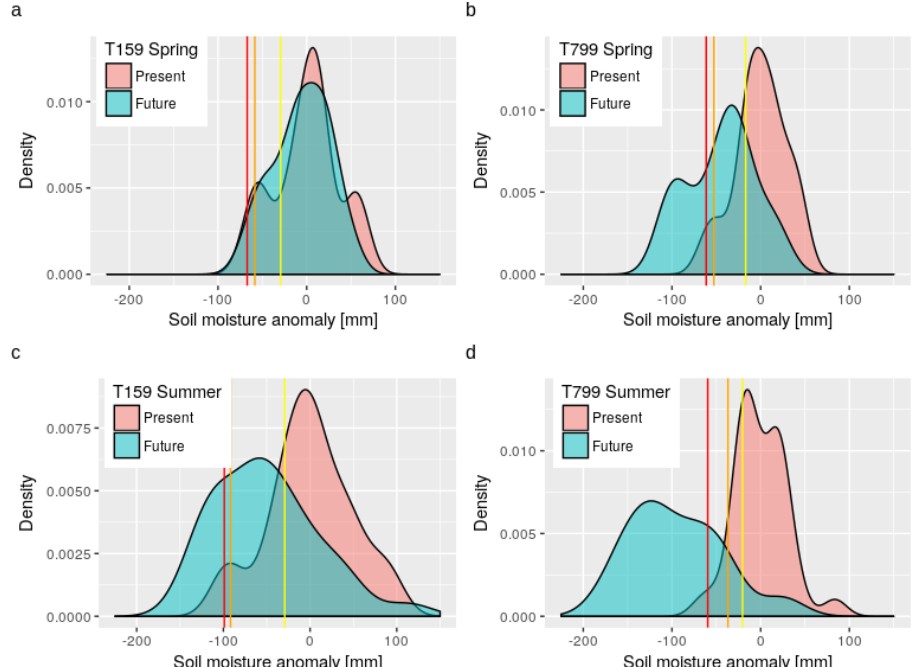

**Figure 4.** Kernel density distribution of soil moisture anomalies over central-western Europe in (a) April–June at T159 resolution, (b) April–June at T799 resolution, (c) July–September at T159 resolution, and (d) July–September at T799 resolution for the present (blue) and future (pink) climates. Extreme (red line), severe (orange line), and moderate (yellow line) drought thresholds are based on the 1st, 5th, and 20th percentile of the present-day soil moisture anomaly distributions.




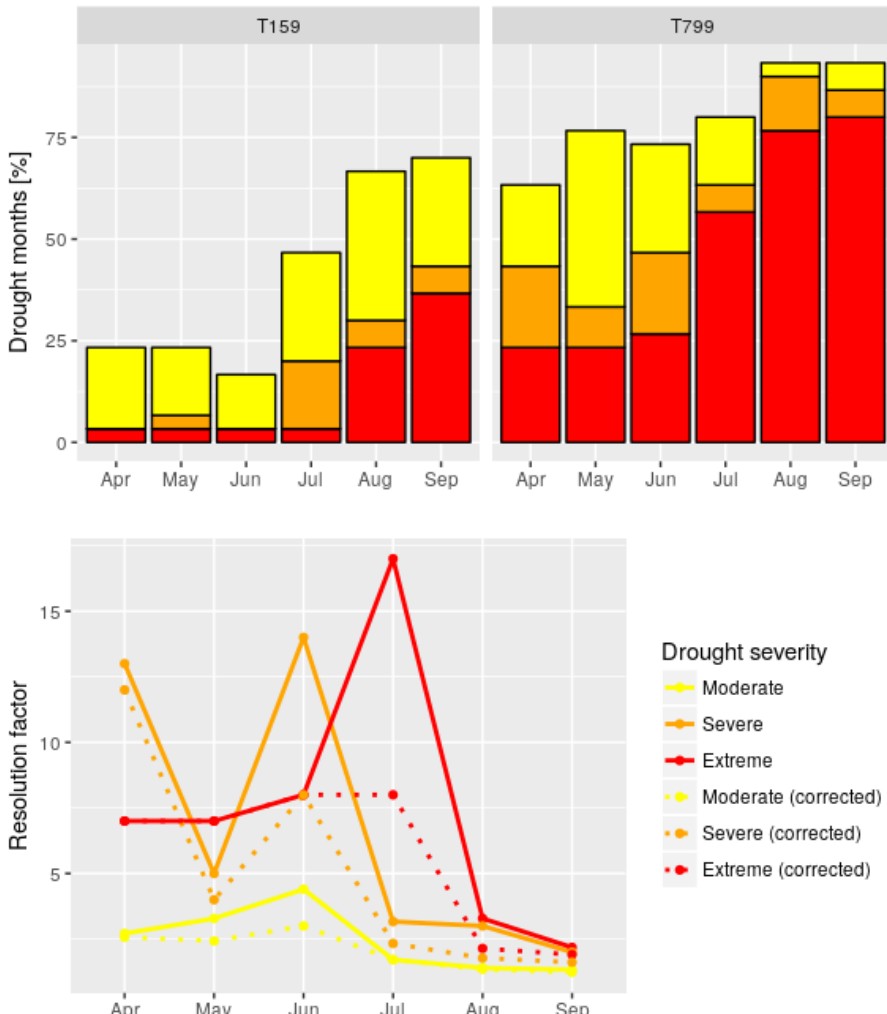

**Figure 5.** Top: Frequency of future droughts, expressed as the percentage of months in extreme (red), severe (orange), and moderate (yellow) drought for (left) T159 and (right) T799 over central-western Europe. Bottom: resolution factor and corrected resolution factor for extreme (red), severe (orange), and moderate (yellow) droughts. The resolution factor (RF) indicates what the impact is of resolution and is defined as the ratio of the frequency of droughts in T799 to the frequency of droughts in T159. The corrected resolution factor (CRF) corrects for shifts in the baseline climate due to higher spatial resolution. CRF is computed similar to RF, but uses the thresholds of soil moisture anomalies based on the climatology of T159 for drought thresholds in T799.





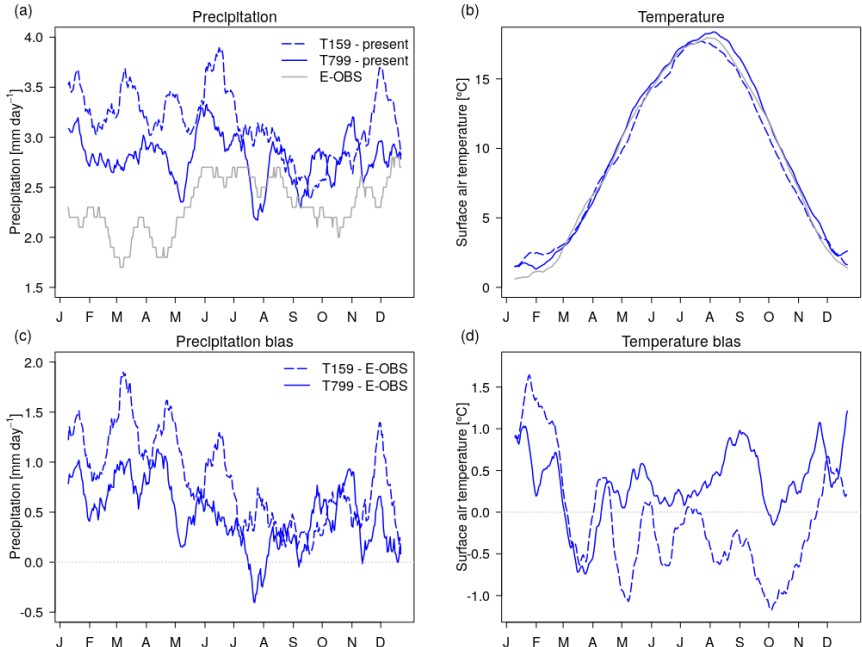

**Figure 6.** Annual cycle of observed (grey) and simulated (blue) (a) precipitation and (b) daily mean near-surface air temperature at T159 (dashed) and T799 (solid) resolution for the present-day climate. Annual cycle of biases of (c) precipitation and (d) daily mean near-surface air temperature with respect to observations. Values are smoothed with a 20-day running mean filter and averaged over central-western Europe. The letters on the horizontal axis correspond with the first day of each month (based on a year with 365 days).



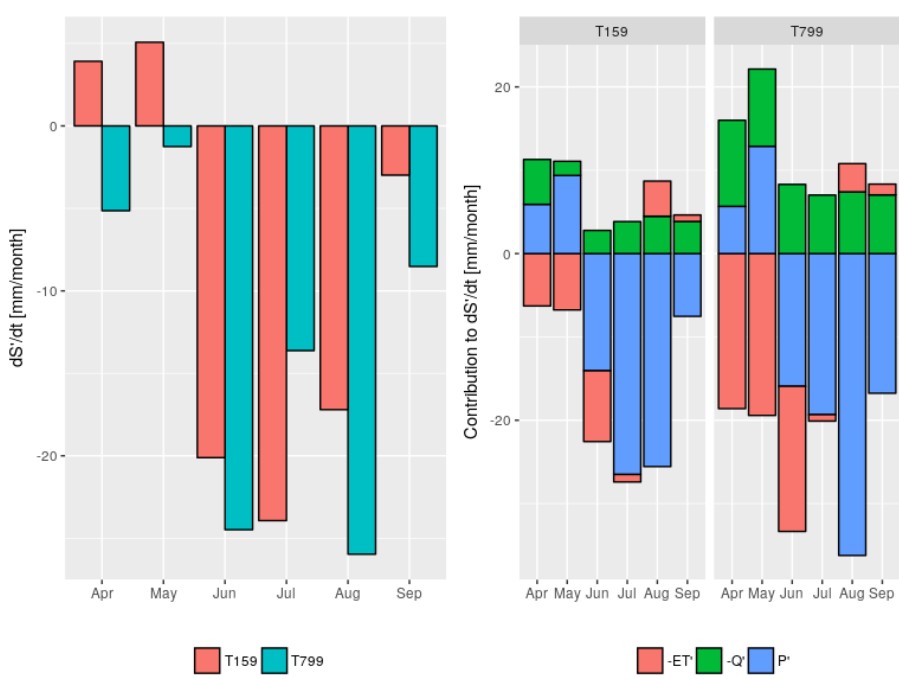

**Figure 7.** Left: Monthly soil moisture storage anomalies (future – present) under future global warming for (pink) T159 and (cyan) T799. Negative is drying, positive is moistening. Right: Contribution of water balance flux anomalies (P', Q', and ET') to soil moisture storage anomalies in the future for (left) T159 and (right) T799. Water balance flux anomalies combine to dS'/dt. Negative anomalies indicate contributions to drying, positive anomalies indicate soil moistening.





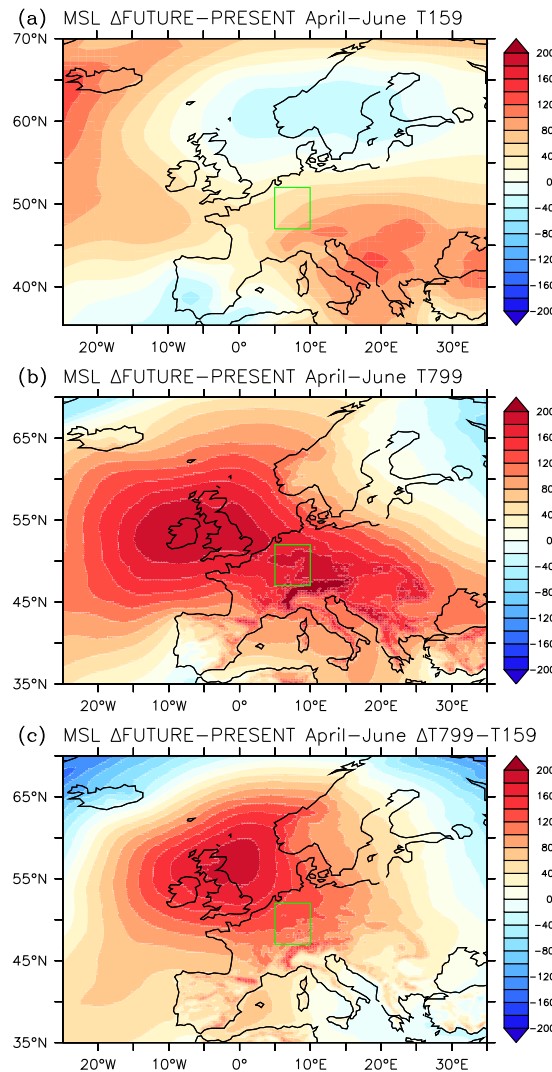

**Figure 8.** Sea level pressure climate change signal (future – present) for spring in the (a) standard and (b) high resolution model. (c) Same but for difference in climate change signal between the high and standard resolution model.





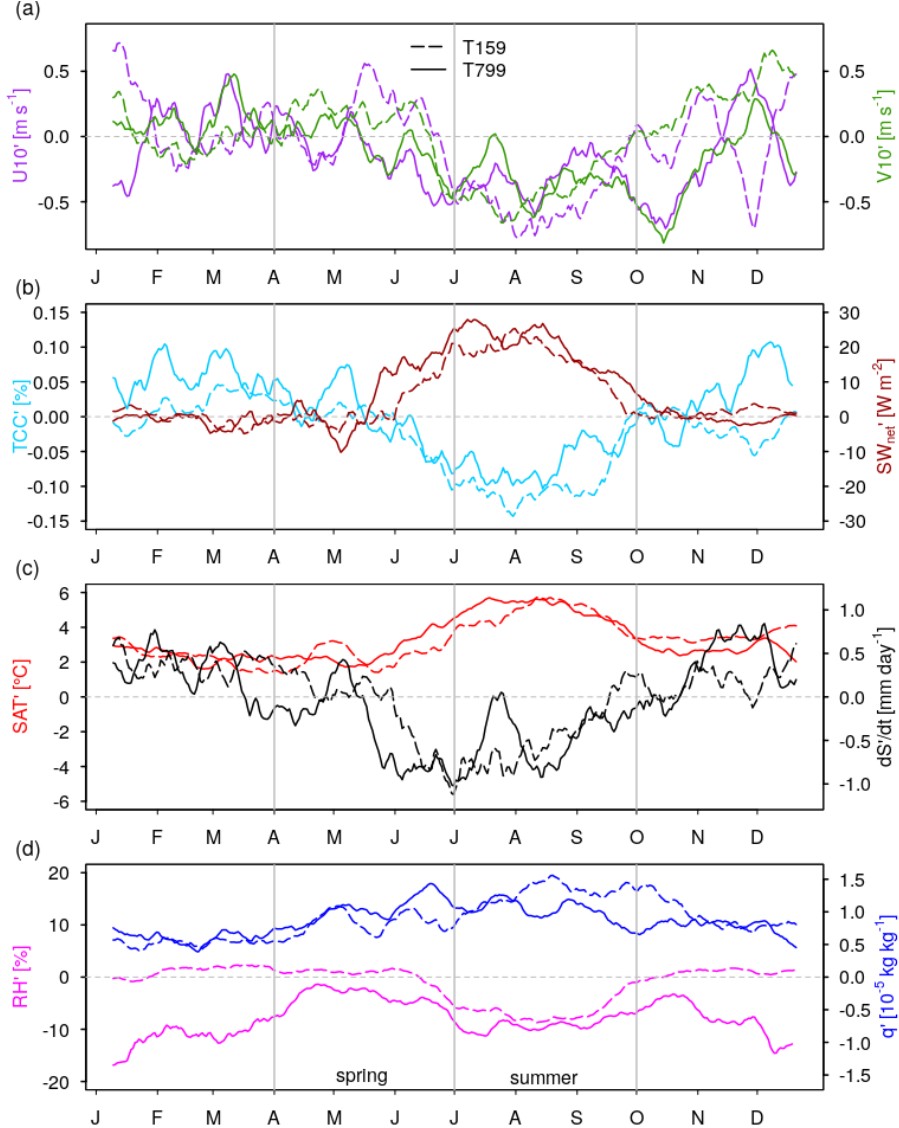

**Figure 9.** Climate change signal (future – present) of (a) zonal (U10') and meridional (V10') wind speed, (b) total cloud cover (TCC') and surface solar radiation (SW$_{net}$'), (c) daily mean surface air temperature (SAT') and soil water storage (dS'/dt), and (d) relative humidty (RH') and 850 hPa specific humidity (q') for T159 (dashed lines) and T799 (solid lines). Values are smoothed with a 20-day running mean filter and averaged over central-western Europe. The letters on the horizontal axis correspond with the first day of each month (based on a year with 365 days).




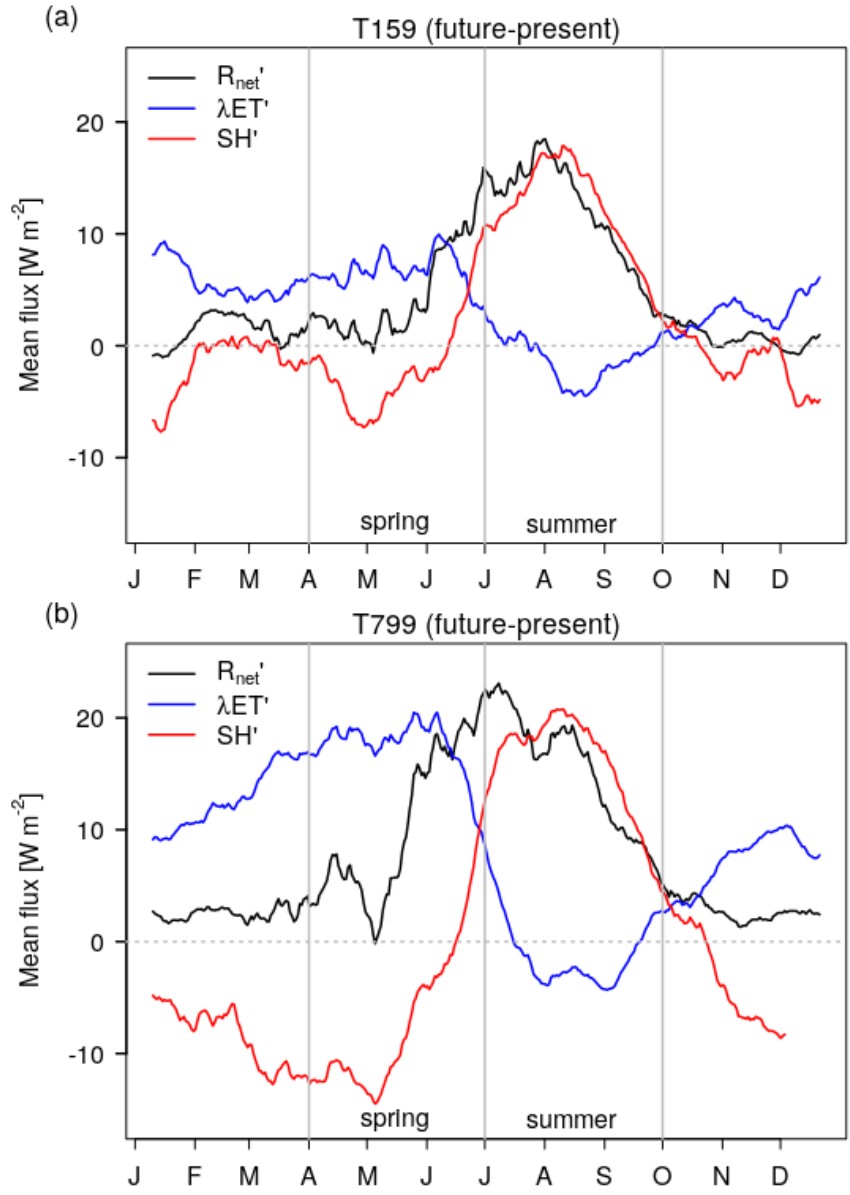

**Figure 10.** Anomalies of surface fluxes under future climate change for (top) T159 and (bottom) T799. Positive indicates enhanced fluxes and negative reduced fluxes. Turbulent sensible heat (SH') and laten heat ($\lambda$ET') fluxes are defined positive upward and net surface radiation ($R_{net}$') is defined positive downward. Values are smoothed with a 20-day running mean filter and averaged over central-western Europe. The letters on the horizontal axis correspond with the first day of each month (based on a year with 365 days).



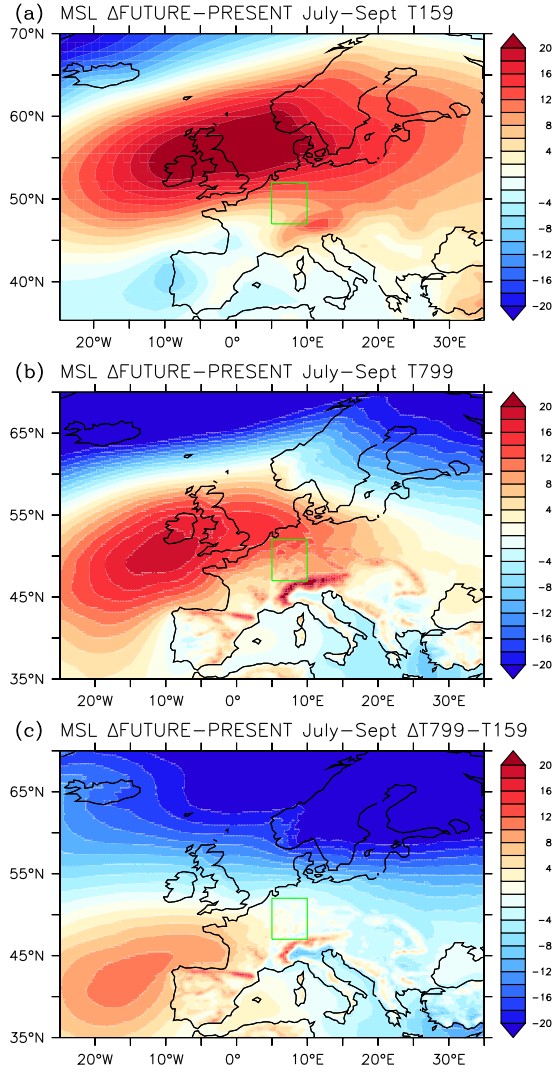

**Figure 11.** Sea level pressure climate change signal (future – present) for summer in the (a) standard and (b) high resolution model. (c) Same but for difference in climate change signal between the high and standard resolution model.



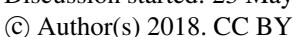

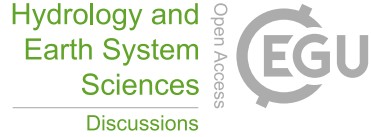

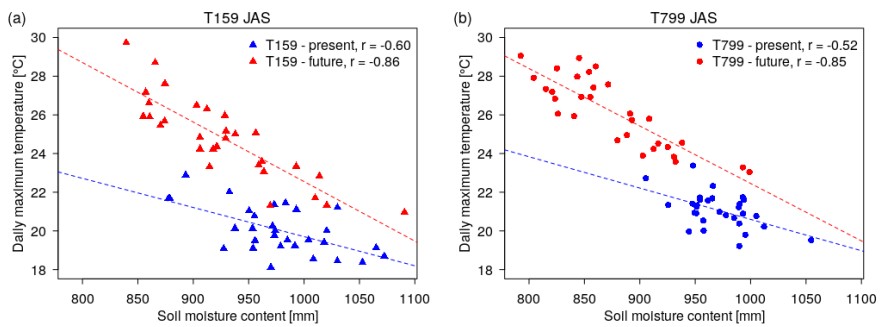

**Figure 12.** Linear regression between summer (July–September) soil moisture content and daily maximum surface air temperature for (a) T159 and (b) T799, for the present (blue) and future (red) climate.





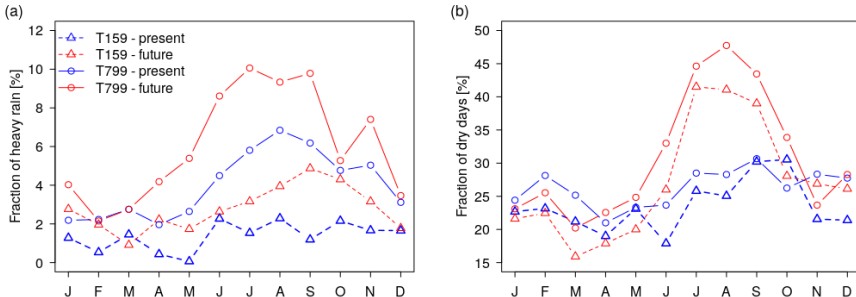

**Figure 13.** (a) Monthly fraction of precipitation falling as heavy rain ($\geq$30 mm day$^{-1}$) in central-western Europe, computed as the contribution of heavy precipitation falling somewhere in the area to the total amount of precipitation. (b) Monthly fraction of dry days averaged over central-western Europe ($\leq$1 mm day$^{-1}$). Before the analysis we regridded precipitation values to the T159 model grid and computed the daily sum of precipitation per gridpoint.