# Peer review of "Impact of climate model resolution on soil moisture projections in central-western Europe"

_Hydrology and Earth System Sciences, 2018_

## Referee Comment (RC1) · Anonymous Referee #1 · 3 Jul 2018

The manuscript entitled "Resolution-dependence of future European soil moisture droughts" by Eveline C. van der Linden presents differences between low (standard) resolution and high-resolution runs of EC-Earth with respect to drought conditions over central Europe. They find that droughts happen to be more severe and durable in the high-resolution experiments and explore potential causes leading to the differences between the model runs.

The manuscript is generally in a good shape, mostly well structured and well written. The overall presentation of the results is good with mostly concise and high quality figures. The methodological approach is well explained and technically sound, but some clarifications are needed in this context. However, the considered ensemble is relatively small and some of the conclusions might not be robust. The authors thus

need to discuss some limitations of their approach before final publication.

Major comment:

The ensemble used in this study consists of 6 ensemble members, each containing 5 years of simulation, resulting in a sample size of 30 years. This is mostly fine for assessing the averaging land surface water and energy balance components. However, I doubt the sample size is large enough for assessing droughts. Droughts are extreme events. An extreme drought event is thus defined as a 1/30 year event, which does not correspond to the 1st quantile, simply because the sample size is too small. It would be good if the authors could discuss the robustness of their results and provide a concise reasoning why the ensemble is not larger. Please also provide some insights into why you choose the years 2002-2006 for present day climate conditions. Maybe because these are the last yers of the CMIP5 historical runs? I was just wondering because in 2003 central Europe experienced a major drought and heatwave.

More comments:

Introduction: It might be good to add a few more references in the first part of the introduction. Especially regarding the uncertainty in European drought trends, such as e.g. Samaniego et al., 2018 (DOI: 10.1038/s41558-018-0138-5)

p. 1, l. 22-25: You write that potential evaporation is enhanced through larger atmospheric moisture demand due to the increasing temperatures. You also write that humidity and wind speed might affect evapotranspiration. This is all a bit confusing, since the atmospheric moisture demand is also defined through humidity and wind speed. Maybe consider to rephrase this part.

p. 2, l. 3: What do you mean by hydroclimatic components?

P. 2, l. 12-14: This is also a bit confusing. You write about quantifying drought severity, and later drought characteristics (such as e.g., severity). Seems redundant.

p. 2, l. 18: Please outline what variables are needed to compute PDSI.

p. 2, l. 26-27: The north-south wetting vs. drying pattern in Europe is actually a well-known feature which was assessed in many studies.

Sec. 2.1 and 2.2: Why do you choose the years 2002-2006? How do the model runs differ? Why don't you use more recent SST data? What version of HTESSEL do you use? Does HTESSEL include river routing or where does the runoff go? Are there open water bodies in HTESSEL?

p. 5, eq. 1: Do you really mean d$\theta$ within the integral? Shouldn't it just be $\theta$?

p. 8, l. 15: The validation period is 1982-2011, right? Might be worth to mention this here as well. What happens if you choose just 2002-2006 as validation period? How is an event like 2003 represented in the model?

p. 9, eq. 4: Well, dS/dt is not necessarily just soil moisture. This might include also snow/ice water storage and water in open water bodies. How is this represented in HTESSEL?

p. 9, l.10-11: Does the soil water content determines runoff?

p. 9, l. 23-27: Here it would help if you could provide more information on how runoff is treated in HTESSEL.

p. 12, l. 19: remove "is"

---

## Referee Comment (RC2) · L. Samaniego (Referee) · 18 Aug 2018

helvet

**Comments on "Resolution-dependence of future European soil moisture droughts"**

(Manuscript # hess-2018-226 )

by E. van der Linden et al.

August 18, 2018

**1  General Comments**

In this manuscript. the authors use a high resolution 6-member ensemble of the general circulation model EC-Earth to get a better and more realistic position of the storm track, which in turn leads to improved representation of the soil moisture (SM) conditions in the future and characterization of SM droughts. The study domain is the central-western Europe under the RCP4.5 emission scenario. One of the major claims of the authors is that high resolution CMIP5 GCMs leads to an underestimation of soil droughts characteristics.

The subject covered by this paper is a highly relevant research topic for practitioners and researchers in hydro-climatology and climate change impacts. I wellcome this study because I am convinced that high resolution GCMs will improve the estimates of future precipitation and temperature patterns because of better parameterization of convective precipitation and landatmosphere feedbacks.

In the present state of the manuscript, nevertheless, there are many shortcomings that have to be clarified before publication.

**2   Specific Comments**

The following technical shortcomings should be addressed in the revised manuscript:

- The literature of future soil moisture drought projections should be updated and the insights of these studies should be put in context of this interesting study. I recommend to include:

  Samaniego, L., S. Thober, R. Kumar, N. Wanders, O. Rakovec, M. Pan, M. Zink, J. Sheffield, E. F. Wood, and A. Marx (2018), Anthropogenic warming exacerbates European soil moisture droughts, Nature Climate Change, 5, 1117–21, doi:10.1038/s41558-018-0138-5.

  Hanel, M., O. X. I. Rakovec, Y. Markonis, P. M. X. ca, L. Samaniego, J. Kysely, and R. Kumar (2018), Revisiting the recent European droughts from a long-term perspective, Scientific Reports, 8(1), 1–11, doi:10.1038/s41598-018-27464-4.

  Hirschi, M. et al. Observational evidence for soil-moisture impact on hot extremes in southeastern Europe. Nat. Geosci. 4, 17–21 (2010).

  Huang, J., Yu, H., Guan, X., Wang, G. & Guo, R. Accelerated dryland expansion under climate change. Nat. Clim. Change 316, 847–171 (2015).

  Berg, A., Sheield, J. & Milly, P. C. D. Divergent surface and total soil moisture projections under global warming. Geophys. Res. Lett. 44, 236–244 (2017).

  ... and references therein.
- L7, P4. Parametric uncertainty plays a very strong role in soil moisture predictions and corresponding drought characteristics (see Samaniego et al, JHM, 2013). For this reason, I consider that a ensemble of 6 members and a single land surface model is too small an ensemble to provide conclusive evidence.

- L17 ff P2: Please clarify in the revised manuscript that the PDSI should not be used for climate impact studies because it does not perform well un non-stationary climate. See the explanation provided in the methods section of Samaniego et al. NCC 2018 and in its supplementary information (Fig S8, S9). Contrary to what Sheffield et al. stated in his Nature paper, the reason of the poor performance of PDSI is more likely related to the autoregressive formulation of this index rather than in the temperature-based PET formulation used in the original formulation of PDSI. The text as it written, put in context with these recent insights, is misleading or at least incomplete.

- L30 ff P2: I strongly suggest to avoid comparisons with the PDSI index (see last point) in future projections. EDgE results (http://edge.climate.copernicus.eu), which are based on downscaled CMIP5 forcings and a multi model ensemble, may be more interesting and realistic than the PDSI estimates. Data is available in nc format upon request (contact L. Samaniego if required).

- L35 ff P2: More recent insights on the future soil moisture droughts can be found in Samaniego et al, NCC 2018, e.g., an increase drought area by $40 \pm 24\%$ by an increase of 3 K. This study also offers a regional perspective that can be put in contrast with the present study.

- L9 P4 Why only the RCP4.5 is used in this study? In my opinion RCP6.0 or 8.5 would be more interesting in the context of future impacts.

- L9, P5 I strongly sugest to use the soil moisture index (see Samaniego et al. JHM 2013, code written in Fortran, it is open source) instead of a soil moisture anomaly. The advantage of SMI is that the SM is mapped to a 0-1 space that allows comparison over time and

space. It facilitates the calculation of drought area, duration and magnitude as presented in Samaniego et al. JHM 2013, Vidal et al. HESS 2010, Andreadis et al. JHM 2005, Sheffield et al. JGR, 2004). The index used in eq.3 is difficult to put in context with past studies.

- L10 P7. Please estimate the severity as used in literature (see previos references). Very interesting will be the changes of the curve area-severity relationship with the resolution of the GCM. The code to estimate this curve as presented in Samaniego et al. JHM 2013 is open source.

- L5 P9. The term "anomaly" as defined in this paragraph is misleading. It is an average change over the domain. I recommend to estimate the change is aridity as defined in Samaniego et al. NCC, 2018 since it is a better estimate of the changes in soil moisture under extreme conditions (droughts). A similar index can be develop to wetter events (just the oposite of the distribution function). I recommend to estimate changes over natural regions to avoid compensation. Some regions experience increases in wetting (Scandinavia), others the oposite (Mediterranean).

- L22 P5, the selection of percentiles is a bit ad-hoc. Why not round numbers like 1, 2, 5, 10, 90, 95, 99 percentiles. Remaning analysis should be updated.

- L11 P14, This hypothesis is highly interesting and should be done as proposed in the future. In this study, however, authors should compare the results existing CMIP5 models (e.g., based on EDgE data ) to see if the hypothesis holds with present insights (see above).

Based on the comments mentioned above and bearing in mind the HESS publishing standards for a research article, I recommend to return it to the authors for major revisions.

L. Samaniego

---

## Author Comment (AC1) · 11 Sep 2018

a ) The manuscript entitled "Resolution-dependence of future European soil moisture droughts" by Eveline C. van der Linden presents differences between low (standard) resolution and high-resolution runs of EC-Earth with respect to drought conditions over central Europe. They find that droughts happen to be more severe and durable in the high-resolution experiments and explore potential causes leading to the differences between the model runs.

The manuscript is generally in a good shape, mostly well structured and well written. The overall presentation of the results is good with mostly concise and high quality figures. The methodological approach is well explained and technically sound, but some clarifications are needed in this context. However, the considered ensemble is relatively small and some of the conclusions might not be robust. The authors thus need to discuss some limitations of their approach before final publication.

> We wish to thank the reviewer for this positive evaluation, and for the constructive comments and insightful suggestions on our paper. They have helped us to substantially improve the quality of the manuscript. Our detailed responses to the comments are presented below.

**Major comment**

b ) The ensemble used in this study consists of 6 ensemble members, each containing 5 years of simulation, resulting in a sample size of 30 years. This is mostly fine for assessing the averaging land surface water and energy balance components. However, I doubt the sample size is large enough for assessing droughts. Droughts are extreme events. An extreme drought event is thus defined as a 1/30 year event, which does not correspond to the 1st quantile, simply because the sample size is too small. It would be good if the authors could discuss the robustness of their results and provide a concise reasoning why the ensemble is not larger. Please also provide some insights into why you choose the years 2002-2006 for present day climate conditions. Maybe because these are the last yers of the CMIP5 historical runs? I was just wondering because in 2003 central Europe experienced a major drought and heatwave.

> Thank you for these questions. We agree that, ideally, extreme drought events as discussed in this paper should be studied with longer model runs. Unfortunately, there are currently no longer runs at this high resolution available for EC-Earth and we currently do not have the resources to increase the ensemble size. This experiment with exceptionally high spatial resolution for a global climate model is computationally very expensive and was therefore performed for multiple research questions. The larger ensemble approach with shorter runs was motivated by a study on the impact of climate change on teleconnection responses to specific tropical SST patterns (Haarsma et al.

2013). As you suggest, the focus of the current paper should therefore be on average changes in the land surface water and energy balance components and on the underlying physical processes rather than on extreme drought events. Therefore we will shift the focus from soil moisture droughts, which are extreme events by definition, to mean changes in soil drying and the associated physical processes, for which 30 years are sufficient. In the revised paper, we will change the title and text accordingly. We will discuss the impact of soil drying on droughts shortly as an impact in the final section of the current paper and we will add a note in which we clarify that a larger sample size is required to obtain robust answers.

Years 2002-2006 are indeed chosen for present-day conditions since these are the last years of the CMIP5 historical runs. This choice was made for multiple research questions and therefore did not consider the major drought and heatwave of 2003. Since this is a model ensemble with perturbed initial conditions, there will not be a major heatwave in each 2003 ensemble member though.

**More comments**

c ) Introduction: It might be good to add a few more references in the first part of the introduction. Especially regarding the uncertainty in European drought trends, such as e.g. Samaniego et al., 2018 (DOI: 10.1038/s41558-018-0138-5)

We will add more references in the first part of the introduction, including your suggestion.

d ) p. 1, l. 22-25: You write that potential evaporation is enhanced through larger atmospheric moisture demand due to the increasing temperatures. You also write that humidity and wind speed might affect evapotranspiration. This is all a bit confusing, since the atmospheric moisture demand is also defined through humidity and wind speed. Maybe consider to rephrase this part.

Thank you for this comment. We understand that this part could be a bit confusing. In the first part of this paragraph we focus on global mean changes (higher temperatures and the associated increase in saturation vapour pressure), whereas the second part focuses on regional changes such as wind speed and associated moisture transports. We will rephrase this paragraph to make the clear distinction between global mean and regional effects.

e ) p. 2, l. 3: What do you mean by hydroclimatic components?

Precipitation and evaporation. We will clarify this in the text.

f ) P. 2, l. 12-14: This is also a bit confusing. You write about quantifying drought severity, and later drought characteristics (such as e.g., severity). Seems redundant.

Thank you for pointing this out. We will remove the redundancy.

g ) p. 2, l. 18: Please outline what variables are needed to compute PDSI.

Following the suggestions of the second reviewer, we have decided to limit the discussion on the PDSI and other off-line drought metrics and we will concentrate more on studies of actual soil moisture changes. The introduction paragraph focusing on PDSI will therefore be replaced.

h ) p. 2, l. 26-27: The north-south wetting vs. drying pattern in Europe is actually a well-known feature which was assessed in many studies.

We agree that this large-scale pattern has been shown in many studies. However, our point is that the magnitude of these changes is highly uncertain and that over many land areas in the transition regions between north and south it is even uncertain whether there will be a wetting or drying trend. Regionally, there are still large uncerainties. We will rephrase this part to avoid confusion.

i ) Sec. 2.1 and 2.2: Why do you choose the years 2002-2006? How do the model runs differ? Why dont you use more recent SST data? What version of HTESSEL do you use? Does HTESSEL include river routing or where does the runoff go? Are there open water bodies in HTESSEL?

We choose years 2002-2006 because these are the final years of the CMIP5 historical runs. Since the individual ensemble members have perturbed initial conditions, the exact years are not of major relevance. We will explain this more extensively in the text.
The model experiments were performed about five years ago, therefore the SST data are not the most recent ones. However, our purpose is to demonstrate the effect of model resolution on future soil moisture changes. To demonstrate this effect these experiments were suitable.
Open water bodies in HTESSEL are represented by open (or frozen) water tiles in the land surface scheme, as described in the data and methods section. HTESSEL does not include river routing. For each grid cell, runoff is transferred to a designated region in the ocean.
We will add these explanations to the 'data and methods' section.

j ) p. 5, eq. 1: Do you really mean d$\theta$ within the integral? Shouldnt it just be $\theta$?

> Thank you for noticing this mistake. We will correct it in the revised manuscript.

k ) p. 8, l. 15: The validation period is 1982-2011, right? Might be worth to mention this here as well. What happens if you choose just 2002-2006 as validation period? How is an event like 2003 represented in the model?

> Thank you for these questions. We understand that this period could be confusing. We will add the section on model validation to the observations description, since this is what the observations are used for. Please note that we use an ensemble of model simulations with perturbed initial conditions (we will explain this in the experimental setup). This indicates a 'present-day'-like forcing similar to 2002-2006, but due to the perturbed initial conditions these years are (and should) not be exactly the same as the observed 2002-2006 conditions. The initial conditions create climate variability, so the simulated 2002-2006 years are not supposed to be exactly the same as the observed 2002-2006 state. Moreover, using only 2002-2006 as a validation period is not statistically robust, since the period only represents five years. The impact of internal climate variability on such a time scale is large, both in the model and in observations. To compare the climatology, you need to compare about 30 years of data. That is why we use a validation period of 30 years for the observations as well. We will explain this in the text.

l ) p. 9, eq. 4: Well, dS/dt is not necessarily just soil moisture. This might include also snow/ice water storage and water in open water bodies. How is this represented in HTESSEL?

> In HTESSEL, a grid box is either 100% land or 100% sea. Each non-land point (grid point with less than 0.5 land cover) can have two fractions, open and frozen water. Open water bodies are thus not included in dS/dt, which only takes land grid points into account. Each land point (grid point with 0.5 or more land cover) can have six fractions of which two include snow (snow on low vegetation/bare ground and high vegetation with snow underneath). Snow is treated as an additional surface layer on top of the upper soil layer. You are therefore right that snow and ice (permafrost) are part of the soil water S in this context and a melt term should be added. However, since we focus on the warm season months and choose our study region outside mountainous regions, ice and snow can be neglected in the land surface water balance. We will add this explanation in the text.

m ) p. 9, l.10-11: Does the soil water content determines runoff?

We understand that this sentence implies a causal relation between soil water content and runoff, which is only partly correct (as described under the next comment) since it is rather a negative feedback. We will rephrase this paragraph.

n ) p. 9, l. 23-27: Here it would help if you could provide more information on how runoff is treated in HTESSEL.

Runoff consists of two parts: surface runoff and subsurface runoff. Whenever precipitation or snow melt occurs a fraction of the water is removed as surface runoff. The ratio runoff/precipitation scales with the standard deviation of orography, and depends on the complexity represented in the gridbox, as well as on soil texture and soil water content. Subsurface runoff is associated with free drainage at the bottom. We will add this information in the description of the land surface scheme and will rephrase these sentences accordingly.

o ) p. 12, l. 19: remove "is"

Thank you, we will correct this sentence.

---

## Author Comment (AC2) · 11 Sep 2018

**L. Samaniego (Referee #2)**

**General comments**

a ) In this manuscript. the authors use a high resolution 6-member ensemble of the general circulation model EC-Earth to get a better and more realistic position of the storm track, which in turn leads to improved representation of the soil moisture (SM) conditions in the future and characterization of SM droughts. The study domain is the central-western Europe under the RCP4.5 emission scenario. One of the major claims of the authors is that high resolution CMIP 5 GCMs leads to an underestimation of soil droughts characteristics.

The subject covered by this paper is a highly relevant research topic for practitioners and researchers in hydro-climatology and climate change impacts. I wellcome this study because I am convinced that high resolution GCMs will improve the estimates of future precipitation and temperature patterns because of better parameterization of convective precipitation and land-atmosphere feedbacks.

In the present state of the manuscript, nevertheless, there are many shortcomings that have to be clarified before publication.

> We thank L. Samaniego for his positive evaluation and the constructive and valuable comments concerning our manuscript, which helped to considerably improve the quality of the paper. We have studied the comments carefully and made major corrections. Our detailed responses to the comments are presented below.

**Specific comments**

The following technical shortcomings should be addressed in the revised manuscript:

b ) The literature of future soil moisture drought projections should be updated and the insights of these studies should be put in context of this interesting study. I recommend to include:

- Samaniego, L., S. Thober, R. Kumar, N. Wanders, O. Rakovec, M. Pan, M. Zink, J. Sheffield, E. F. Wood, and A. Marx (2018), Anthropogenic warming exacerbates European soil moisture droughts, Nature Climate Change, 5, 111721, doi:10.1038/s41558- 018-0138-5.

- Hanel, M., O. X. I. Rakovec, Y. Markonis, P. M. X. ca, L. Samaniego, J. Kysely, and R. Kumar (2018), Revisiting the recent European droughts from a long-term perspective, Scientific Reports, 8(1), 111, doi:10.1038/s41598-018-27464-4.

- Hirschi, M. et al. Observational evidence for soil-moisture impact on hot extremes in southeastern Europe. Nat. Geosci. 4, 1721 (2010).

- Huang, J., Yu, H., Guan, X., Wang, G. & Guo, R. Accelerated dryland expansion underclimate change. Nat. Clim. Change 316, 847171 (2015).

- Berg, A., Sheield, J.& Milly, P. C. D. Divergent surface and total soil moisture projections under global warming. Geophys. Res. Lett. 44, 236244 (2017).
- ... and references therein.

Thank you for these recommendations. We will update and revise the introduction with these relevant recent insights and add the suggested references in the text.

c ) L7, P4. Parametric uncertainty plays a very strong role in soil moisture predictions and corresponding drought characteristics (see Samaniego et al, JHM, 2013). For this reason, I consider that a ensemble of 6 members and a single land surface model is too small an ensemble to provide conclusive evidence.

We agree that a larger ensemble of different models should be used to obtain robust answers about the impact of model resolution. The use of a single model allowed us to put emphasis on mechanistic explantations for the differences between model resolutions. Certainly, these results should be verified with a larger systematic ensemble of high resolution model simulatons such as the CMIP6-endorsed HighResMIP, which are currently not available. Therefore, we will add notes in the manuscript to emphasize that a larger ensemble is required to obtain conclusive evidence.

d ) L17 ff P2: Please clarify in the revised manuscript that the PDSI should not be used for climate impact studies because it does not perform well un non-stationary climate. See the explanation provided in the methods section of Samaniego et al. NCC 2018 and in its supplementary information (Fig S8, S9). Contrary to what Sheffield et al. stated in his Nature paper, the reason of the poor performance of PDSI is more likely related to the autoregressive formulation of this index rather than in the temperature-based PET formulation used in the original formulation of PDSI. The text as it written, put in context with these recent insights, is misleading or at least incomplete.

Thank you for this insightful suggestion. We will add the explanation provided by Samaniego et al. NCC 2018 in the introduction. In addition, we will strongly reduce the paragraph that discusses the PDSI results and will shift the focus to soil moisture projections instead.

e ) L30 ff P2: I strongly sugest to avoid comparisons with the PDSI index (see last point) in future projections. EDgE results (http://edge.climate.copernicus.eu), which are based on downscaled CMIP5 forcings and a multi model ensemble, may be more interesting and realistic than the PDSI estimates. Data is available in nc format upon request (contact L. Samaniego if required).

We clearly understand your point. We will remove the paragraph in which a comparison of PDSI results is made and we will revise the introduction to shift the focus to actual soil moisture projections instead of drought metrics.

f ) L35 ff P2: More recent insights on the future soil moisture droughts can be found in Samaniego et al, NCC 2018, e.g., an increase drought area by 40 ± 24% by an increase of 3 K. This study also offers a regional perspective that can be put in contrast with the present study.

We will add a regional comparison with this study in the conclusions section.

g ) L9 P4 Why only the RCP4.5 is used in this study? In my opinion RCP6.0 or 8.5 would be more interesting in the context of future impacts.

Although we would have liked to include other RCPs, we did not have the resources to repeat the runs at this very high resolution for other RCPs as well. Therefore, RCP4.5 was chosen since it is one of the middle scenarios. If changes are significant in RCP4.5, they are very likely also significant in the higher RCPs. Our aim is to demonstrate the impact of spatial resolution on future soil drying and to provide a mechanistic explanation. For this goal, one RCP was considered sufficient.

h ) L9, P5 I strongly sugest to use the soil moisture index (see Samaniego et al. JHM 2013, code written in Fortran, it is open source) instead of a soil moisture anomaly. The advantage of SMI is that the SM is mapped to a 0-1 space that allows comparison over time and space. It facilitates the calculation of drought area, duration and magnitude as presented in Samaniego et al. JHM 2013, Vidal et al. HESS 2010, Andreadis et al. JHM 2005, Sheffield et al. JGR, 2004). The index used in eq.3 is difficult to put in context with past studies.

Thank you for this suggestion. We understand your point and agree that SMI (as calculated in Samaniego et. al. JHM 2013) is a good measure for multi-model and regional intercomparison studies. In our study, however, we mainly focus on mean changes in soil drying (aridity) rather than on soil moisture droughts (extremes). In addition, we do not focus on regional patterns but on a regional average value over central-western Europe. From your comment, we have realized that this main focus was not clear enough from the paper as written. Therefore, we will revise the paper accordingly by changing the title and structure of the paper. We will remove the term 'drought' from the title and replace it with 'soil moisture changes'. Furthermore, we will include more specifically the study region 'central-western Europe' rather than 'Europe' in the title. We will also restructure the paper by moving the section on droughts towards the impact section at the end of the paper. The main body of the paper will be clearly

focused on regional average soil moisture changes and its underlying mechanisms. The absolute drying measure (expressed in soil moisture changes) facilitates the quantitative comparison of different water balance components (precipitation, evapotranspiration, runoff, soil moisture storage) in terms of magnitude.

i ) L10 P7. Please estimate the severity as used in literature (see previous references). Very interesting will be the changes of the curve area-severity relationship with the resolution of the GCM. The code to estimate this curve as presented in Samaniego et al. JHM 2013 is open source.

Thank you for this suggestion. We have computed the severity based on percentile thresholds of the soil moisture anomaly distribution, as is done in for example Burke (2007, JHM) and Zhao (2015, J. Clim.) for CMIP5 models. The results show that broad change pattern in EC-Earth is similar to other CMIP5 simulations. We will add this in the text. Considering the area-severity relationship, we believe this would be very interesting. However, we did not concentrate on drought area in our study since we focus on a regional average value over the Rhine-Meuse drainage basin and not on the entire European continent. We believe that your suggestion would be an excellent idea for a study focusing on the continental scale changes, but in our current study we only focus on a relatively small region in central-western Europe. Therefore, we will take your suggestion into account for a future study focussing on the impact of model resolution on future European patterns of droughts with multiple global climate models.

j ) L5 P9. The term anomaly as defined in this paragraph is misleading. It is an average change over the domain. I recommend to estimate the change is aridity as defined in Samaniego et al. NCC, 2018 since it is a better estimate of the changes in soil moisture under extreme conditions (droughts). A similar index can be develop to wetter events (just the oposite of the distribution function). I recommend to estimate changes over natural regions to avoid compensation. Some regions experience increases in wetting (Scandinavia), others the oposite (Mediterranean).

We understand that this terminology could be a bit misleading, therefore we will not call future mean changes in soil moisture or other water balance components 'anomalies'. We will change this terminology throughout the text.
As mentioned before, we believe that there was some confusion in the text about 'droughts' (extreme conditions) and 'drying' (mean changes). The main focus of our paper is not on extreme conditions but on mean future changes. Where appropriate, we will clarify this in the text.
Our focus region is roughly over the Rhine-Meuse drainage basin in central-western Europe, which is a natural region. In Figure 2 we show that the changes in aridity over this region are not compensating but have the same sign.

k ) L22 P5, the selection of percentiles is a bit ad-hoc. Why not round numbers like 1, 2, 5, 10, 90, 95, 99 percentiles. Remaning analysis should be updated.

We understand that these percentiles could be confusing, since they are indeed not round numbers. This is because we only have 30 years of data. Therefore we can only compute e.g. 1/30, 2/30 or 6/30 year events, which we classified as extreme, severe, and moderate droughts, respectively. To avoid confusion we have rewritten this paragraph to make this clear. In addition, we have moved this subsection to the final part of the paper which focuses on extremes rather than mean changes in soil moisture.

l ) L11 P14, This hypothesis is highly interesting and should be done as proposed in the future. In this study, however, authors should compare the results existing CMIP5 models (e.g., based on EDgE data ) to see if the hypothesis holds with present insights (see above).

Thank you for this suggestion. Unfortunately, we do not (yet) have future simulations of other high resolution models to test our hypothesis at the moment, since the model runs were performed with only one global climate model. Existing CMIP5 models do not have this high spatial resolution. In fact, EC-Earth in its 'standard' CMIP5 resolution has already a high spatial resolution compared to other CMIP5 models. In a future study, in which we will have systematic model simulations with multiple high resolution GCMs, we will be able to make this comparison.

---

## Author Response (AR2)

a ) This is a re-review of the manuscript entitled "Impact of climate model resolution on soil moisture projections in central-western Europe" by Eveline C. van der Linden et al. presenting differences between low (standard) resolution and high-resolution runs of EC-Earth with respect to soil moisture over central Europe. Compared to the previous version, the authors have changed the main focus of the study from assessing soil moisture droughts to mean changes in soil moisture conditions.

I appreciate the changes made by the authors and their thorough response to the reviewer's comments. The manuscript is now better structured and the modeling approach is explained in more detail. However, I still query the robustness of the drought assessment and would like the authors to enhance the discussion on flaws and limitations of their approach (see comment below). Apart from this, it is my assessment that the manuscript is suitable for publication in HESS after considering the additional comments provided below.

> We wish to thank the reviewer for the positive evaluation and constructive comments. We have made the requested corrections in the manuscript as indicated in tracked changes throughout the text.

Additional comments (page and line numbers refer to the tracked changes document provided in the author response):

b ) p. 1, l. 24: See e.g., also Greve et al. (2017, https://doi.org/10.1088/1748-9326/aa89a3) for a more in-depth soil moisture analysis over a wide range of climates. This might also be referenced on p. 3, l. 8, adding also Scheff et al. (2017, https://doi.org/10.1175/JCLI-D-16-0854.1). Both studies provide a direct assessment of climate model output.

> Thank you for these suggestions. We have added these references in the text.

c ) p. 6, l. 16: This might be confusing to many readers. You better write "aridity metric" or "measure of aridity". The "aridity index" is usually referred to as the ratio of potential evaporation and precipitation.

> Thank you for pointing this out. We changed "index" to "metric".

d ) p. 6, l.24ff: You have a limited sample size, which is not a problem per se, but does to a certain extent limit the robustness of your results regarding droughts. I do not want you to remove the drought assessment from the paper. However, I would really encourage you to include a discussion on the limitations of your approach using a rather small sample to define drought severity. A 1/30yrs event might happen 4 times in another 30 yrs sample. Or it never happens. Hence, a separation into extreme, severe and moderate droughts seems a bit random to me and you may think about using only one drought class? If you keep the 3 classes, please discuss the potential flaws of this approach in more detail. You might also consider to avoid providing percentage values here. A definition based on an x/30yrs is sufficient in my opinion (like you do in Sec. 5.1).

We agree that the sample size is limited and three drought classes are therefore too specific. Following your suggestion, we have changed the number of classes to one drought class (previously the moderate class), which is the 20th percentile category corresponding to 6/30 year events. This makes the results more robust. We have changed the text in section 5.1 and Figures 10 and 11 accordingly. Furthermore, in the last sentence of the abstract, section 5.1 and section 6 we emphasised that our results are only based on one model and 30 years of data, and therefore futher research with a multimodel high resolution ensemble is required to make more solid conclusions about future drought changes.

e ) p. 7, l. 11: "European distribution of future soil moisture anomalies" sounds a bit confusing here (I would expect something like in Fig. 10). Better write "maps of future soil moisture anomalies"

We agree. We have changed it in the text.

f ) p. 10, l. 10ff: I appreciate the clarifications you made regarding the storage term in the land water balance. However, please consider being consistent with the terms "soil water storage", "soil water content", "soil moisture", and "soil moisture storage". Please also note again that ice and snow are the only other relevant storage components in HTESSEL and open water bodies are not included.

We have made the terms consistent using "moisture" throughout the manuscript. We also checked the manuscript for the correct use of "soil moisture" (state) versus "soil moisture storage" (changes over time). Please note that we mentioned in the land surface scheme description (Section 2.2) that open water is not included in the "land tiles" of HTESSEL and therefore not in the land surface water balance.

[revised manuscript text omitted]